# A Comparative Study of the Synthesis and Characterization of Biogenic Selenium Nanoparticles by Two Contrasting Endophytic Selenobacteria

**DOI:** 10.3390/microorganisms11061600

**Published:** 2023-06-16

**Authors:** Eulàlia Sans-Serramitjana, Carla Gallardo-Benavente, Francisco Melo, José M. Pérez-Donoso, Cornelia Rumpel, Patricio Javier Barra, Paola Durán, María de La Luz Mora

**Affiliations:** 1Center of Plant, Soil Interaction and Natural Resources Biotechnology, Scientific and Biotechnological Bioresource Nucleus (BIOREN-UFRO), Universidad de La Frontera, Avenida Francisco Salazar 01145, P.O. Box 54-D, Temuco 4811230, Chile; patricio.barra@ufrontera.cl (P.J.B.); paola.duran@ufrontera.cl (P.D.); 2Centro Biotecnológico de Estudios Microbianos (CEBEM), Universidad de La Frontera, Temuco 4811230, Chile; carlagallardo6@gmail.com; 3Departamento de Física, Center for Soft Matter Research, SMAT-C, Usach, Avenida Ecuador, Estación Central, Santiago 9170124, Chile; francisco.melo@usach.cl; 4BioNanotechnology and Microbiology Lab, Center for Bioinformatics and Integrative Biology, Facultad de Ciencias de la Vida, Universidad Andres Bello, Santiago 8370133, Chile; jose.perez@unab.cl; 5Institute of Ecology and Environmental Sciences, UMR 7618, CNRS-UPMC-UPEC-INRAE-IRD, Sorbonne University, 75005 Paris, France; cornelia.rumpel@inrae.fr; 6Biocontrol Research Laboratory, Universidad de La Frontera, Temuco 4811230, Chile

**Keywords:** biogenic nanoparticles, selenium, selenite, *Bacillus paranthracis*, *Enterobacter ludwigii*, biofortification

## Abstract

The present study examined the biosynthesis and characterization of selenium nanoparticles (SeNPs) using two contrasting endophytic selenobacteria, one Gram-positive (*Bacillus* sp. E5 identified as *Bacillus paranthracis*) and one Gram-negative (*Enterobacter* sp. EC5.2 identified as *Enterobacter ludwigi*), for further use as biofortifying agents and/or for other biotechnological purposes. We demonstrated that, upon regulating culture conditions and selenite exposure time, both strains were suitable “cell factories” for producing SeNPs (B-SeNPs from *B. paranthracis* and E-SeNPs from *E. ludwigii*) with different properties. Briefly, dynamic light scattering (DLS), transmission electron microscopy (TEM), and atomic force microscopy (AFM) studies revealed that intracellular E-SeNPs (56.23 ± 4.85 nm) were smaller in diameter than B-SeNPs (83.44 ± 2.90 nm) and that both formulations were located in the surrounding medium or bound to the cell wall. AFM images indicated the absence of relevant variations in bacterial volume and shape and revealed the existence of layers of peptidoglycan surrounding the bacterial cell wall under the conditions of biosynthesis, particularly in the case of *B. paranthracis*. Raman spectroscopy, Fourier transform infrared spectroscopy (FTIR), energy-dispersive X-ray (EDS), X-ray diffraction (XRD), and X-ray photoelectron spectroscopy (XPS) showed that SeNPs were surrounded by the proteins, lipids, and polysaccharides of bacterial cells and that the numbers of the functional groups present in B-SeNPs were higher than in E-SeNPs. Thus, considering that these findings support the suitability of these two endophytic stains as potential biocatalysts to produce high-quality Se-based nanoparticles, our future efforts must be focused on the evaluation of their bioactivity, as well as on the determination of how the different features of each SeNP modulate their biological action and their stability.

## 1. Introduction

Selenium (Se) is an essential micronutrient with beneficial effects for human health at low concentrations due to its antioxidant, anticancer, antimicrobial, and immunoregulatory properties [1,2]. The acquisition of optimal Se levels from the daily diet can promote a reduction in many pathological disorders, such as reproductive disorders, diabetes, thyroid issues, cancer, and immune responses [2,3]. The primary dietary Se sources are the edible parts of crop plants, while the Se content in crops is highly dependent on soil Se bioavailability, which is directly influenced by their physical and agrochemical properties [4,5]. Therefore, a mineral imbalance in the soil–plant continuum can lead to Se-deficient food production with consequences for both human and animal nutrition [6]. This represents a concern, especially in Andisols from southern Chile, which represent around 60% of agricultural soils, where the most important milk, meat, and cereal crops are produced [7]. However, these soils are characterized by their low Se content (between 0.02 and 0.18 mg kg^−1^ soil) and present high acidification rates, one of the main factors limiting agricultural production [8,9].

In this context, agronomic biofortification through the application of inorganic Se fertilizers could be considered a potential tool to raise Se levels in the human diet. However, in volcanic Andisols from Chile, several studies have shown that agronomic biofortification is inadequate, since Se can form stable complexes with clays and/or can be strongly adsorbed onto the oxyhydroxides of aluminum (Al), iron (Fe), or manganese (Mn), resulting in low Se bioavailability to plants [8,9,10].

In view of this scenario, over the last few years, our research group has widely investigated the use of rhizospheric and endophytic microorganisms isolated from cereal crops grown in Andisols from southern Chile as a biotechnological strategy for Se biofortification [11,12,13,14,15]. In this regard, it has been observed that the use of endophytic bacteria frequently provides greater benefits to plants than rhizobacteria in relation to plant nutrition, the hormonal regulation of root growth, disease suppression, and Se tolerance [16,17,18,19]. In fact, Durán et al. [12] determined that endophytes are better adapted to elevated Se concentrations than rhizospheric bacteria, thus being the most effective approach for enhancing Se uptake by plants. Moreover, considering that microorganisms play a pivotal role in the Se cycle in nature [20], endophytic selenobacteria have also been proposed as a potential tool for the bioremediation of Se-contaminated soils [21].

Interestingly, Durán et al. [14] showed that plant-growth-promoting (PGP) endophytic selenobacteria grown on Se-added media are able to metabolize the most important organic Se forms, such as SeMet and SeMetSeCys, as well as Se particles (>100 nm). Nonetheless, the authors indicated that due to the large size of Se spheres, they were not able to penetrate inside the plant roots [22], thus limiting their use as biofortifying agents. In fact, many bacteria have been described to have the ability to reduce selenite to elemental Se (Se^0^) [23]. However, the Se^0^ form is more commonly found as Se particles (100–500 nm) than as SeNPs (<100 nm) [23].

SeNPs are generating interest in many areas of science because of their low toxicity and high biocompatibility compared to other forms of the element [24,25,26,27]. In agriculture, SeNPs are mainly used as antimicrobial, pest control, biostimulant [28], biofortifying [29], stress alleviation [30,31], and nutraceutical agents [32]. Biofortification using SeNPs has emerged in recent years as a promising alternative to conventional Se fertilizers for enriching crops [27,33]. The interest in SeNPs for the biofortification of plant food stems from the potential for their slow Se release, avoiding possible losses in agroecosystems when commercial fertilizers are used [29].

SeNPs can be synthesized by biological, physical, and chemical methods [34]. In general, the synthesis of NPs using organisms such as plants, bacteria, fungi, yeasts, and algae is less expensive and safer since it uses eco-friendly non-toxic materials [34]. Moreover, biogenic SeNPs are much more stable than those produced by non-green methods due to the natural coating of organic materials on the surface, which prevents agglomeration and uncontrolled growth [35]. The biosynthesis of SeNPs using bacteria (both anaerobic and aerobic) has been extensively explored due to their ability to colonize, adapt to, and persist in detrimental environmental niches. However, it is of great interest to search for new bacterial strains capable of producing SeNPs in order to establish a more comprehensive data bank and to identify the factors affecting the properties of SeNPs. Indeed, several studies have established that the biological functions of SeNPs depend upon various factors, such as the synthesis method, surface composition, morphology, and size. These characteristics may cause the intake of plant foods biofortified with SeNPs to have a different effect within the human body compared with other sources of Se that are conventionally used.

Amongst the bacterial populations, endophytic bacteria are considered potential candidates for NP bio-production due to their high secretion of active metabolites [36]. Their potential metabolic activity enables them to fabricate NPs with varied sizes, shapes, and high stability [37]. However, at present, there are only a few data on the biogenic synthesis of SeNPs by endophytic selenobacteria [38].

Therefore, based on the above, and given that SeNP biofortification today is proposed as an alternative to conventional Se fertilizers, the aim of this work was to explore the capability of two contrasting endophytic selenobacteria (*B. paranthracis* and *E. ludwigi*) by producing SeNPs and optimizing the synthesis conditions and comparing their physico-chemical and structural features. Here, we report for the first time the use of both endophytic isolates as a sustainable and eco-friendly system to obtain SeNPs with potential impact on human health, agriculture, and biotechnological applications through the improvement of the nutritional quality of the crops with Se, the optimization of the use of natural resources, and the increase in food security by reducing chemical fertilization.

## 2. Materials and Methods

### 2.1. Bacterial Strains and Culturing

The endophytic selenobacteria isolated from roots of wheat plants (*Bacillus* sp. E5 identified as *Bacillus paranthracis* Accession no. KF561868 and *Enterobacter* sp. EC5.2 identified as *Enterobacter ludwigi* Accession no. KF561858) were selected for Se tolerance [13]. Both isolates were routinely pre-cultured aerobically in Luria Bertani (LB) broth (Becton Dickinson and Company, Sparks, MD, USA) at 30 °C with orbital shaking at 120 rpm. When needed, LB medium was solidified by adding 15 g/L of agar (OxoidTM, Thermo Fisher Scientific, Waltham, MA, USA).

### 2.2. SeNP Biosynthesis

The biosynthesis of SeNPs was evaluated in selected isolates following the protocol described by Gallardo et al. [39,40] with a few modifications. Briefly, a bacterial pre-inoculum was diluted on LB medium (1:100) and grown at 30 °C with shaking at 120 rpm for 24 h. Then, cultures were harvested by centrifugation for 10 min at 7000 rpm (Centurion Scientific Pro Analytical CR4000R, Chichester, UK) and washed once with 100 mM Tris-HCl pH 8 (Santa Cruz Biotechnology, Chem Cruz, Dallas, TX, USA). The resulting bacterial pellets were resuspended in 400 mL of 100 mM Tris-HCl pH 8, adjusted to 10^8^ CFU mL^−1^, and challenged with filter-sterilized sodium selenite (Na_2_SeO_3_ × 5H_2_O; Sigma-Aldrich, St. Louis, MO, USA) to a final concentration of 5 mM as per Durán et al. [12]. The mixture was incubated at 30 °C in an orbital shaker (120 rpm) for different time periods (4, 6, 24, and 48 h). Bacterial inocula without the addition of selenite were maintained as control. The reduction of SeO_3_^−2^ to Se^0^ was monitored primarily by changes in the color of bacterial suspensions. The biosynthesis of SeNPs by both *Bacillus* sp. E.5 and *Enterobacter* sp. EC5.2 was also assessed by imaging both bacterial cells grown in the presence of Na_2_SeO_3_ using transmission electron microscopy (TEM) and atomic force microscopy (AFM).

### 2.3. Determination of Total Se in Bacterial Biomass

The quantities of Se accumulated by *Bacillus* sp. E.5 and *Enterobacter* sp. EC5.2 in the presence of selenite were determined according to the methodology described by Kumpulainen et al. [41]. Briefly, the cell pellets of both Se-enriched cultures were collected by centrifugation for 15 min at 6000 rpm and rinsed twice with sterile saline at 0.85%. Then, bacterial pellets were resuspended in a protective medium (10% skim milk—Difco Laboratories, Detroit, Mich.) at a ratio of 1:2 and lyophilized overnight. All samples were weighted and, thereafter, digested in 10 mL of acid mixture (65% HNO_3_, 70% HClO_4_, and 95% H_2_SO_4_) and incubated overnight at room temperature. After incubation, the mixture was heated at 120 °C for 3 h and 220 °C for 5 h, and then HCl (12%) was added in order to reach 15 mL. Finally, the mixture was boiled at 120 °C for 20 min. The Se content was measured via atomic absorption spectrophotometry (AAS) with a Hydride generator (HG) 3000 (GBC Scientific Equipment Ltd, Victoria, Australia) using NaBH_4_ solution as a reducing agent. Two Se-enriched flour samples supplied by the Department of Applied Chemistry and Microbiology of Helsinki University (Finland) were used as references.

### 2.4. TEM and AFM Imaging

Bacterial cultures grown with and without 5 mM Na_2_SeO_3_ were harvested by gentle centrifugation at 7000 rpm for 5 min. The pellets were then washed three times with deionized water and fixed in a 2.5% solution of glutaraldehyde in 0.1 M cacodylate buffer pH 7.2 for at least 1 h at room temperature. After fixation, the pellets were rinsed in 0.1 M cacodylate buffer (pH 7.2) three times for 5 min each, and then 5 μL of cells was deposited onto carbon-coated copper grids (CF300-CU, Electron Microscopy Sciences, Hatfield, PA, USA), which were then air-dried prior to visualization using a Zeiss Libra 120 plus TEM at an acceleration voltage of 120 kV.

AFM was also used to visualize both endophytic strains after 6 h exposure to Na_2_SeO_3_ (5 mM). Briefly, 100 μL of bacterial suspensions was deposited and incubated in the dark for 90 min on the surface of positively charged glass coverslips previously coated with a 0.2% (*w*/*v*) Poly(ethyleneimine) (Sigma-Aldrich, Dorset, UK) solution to enhance bacterial cell adhesion and to prevent the removal of the bacteria during experiments. Then, the glass coverslips were rinsed three times in Tris buffer to eliminate the excess of cells and left to air dry for 15 min. Samples were imaged in air using AFM (Witec Alpha 300, Ulm, Germany). All images were collected in AC mode at room temperature using Hi’Res-C14/Cr-Au probe of thickness 2.1 μm, length 125 μm, width 25 μm, resonance frequency 160 kHz, and a force constant of 5 N/m. The data acquired during surface scanning were converted into images of topography and amplitude. In the topography images, the shape, structure, and surface of selenite-tolerant bacteria could be observed. Furthermore, amplitude images allow the visualization of fine surface details of bacterial cells [42].

### 2.5. Purification of Biosynthesized SeNPs

SeNPs were separated from the culture following a published protocol with certain modifications [40]. After the incubation period, bacterial cultures (50 mL) were centrifuged for 15 min at 6000 rpm. Supernatants containing extracellular SeNPs were filtered through 0.2 µm filters to remove cellular debris. After the removal of the supernatants, the pellets containing intracellular SeNPs were washed once with deionized water and resuspended for sonication. The pellets were disrupted through ultrasonication on ice (ten cycles of 30 s of sonication with 30 s of rest). After this, the suspensions were centrifuged, followed by 30 min centrifugation at 6000 rpm to separate unbroken cells. Then, both supernatants containing intra- or extracellular SeNPs were collected and filtered through 0.22 μm filters (BioLab) to remove cellular debris. Filtered supernatants were then concentrated using a 10 kDa membrane (Amicon Ultra-15 tube, Merck Millipore Ltd., Dublin, Ireland). Concentrated SeNP suspensions were lyophilized overnight and stored at 4 °C until use.

### 2.6. Characterization of SeNPs

#### Dynamic Light Scattering Measurements (DLS)

The hydrodynamic size, zeta potential, and polydispersity index (PDI, which is an indicator of size heterogeneity) of SeNPs suspended in ultra-pure water were measured in a Zetasizer Nano ZS (Malvern Instruments Co., Malvern, UK). Polystyrene cuvettes with a path length of 10 mm at 25 °C and a refractive index of 1.6 were used. All measurements were performed in triplicate.

### 2.7. TEM

TEM analysis was performed to characterize the size and shape of synthesized SeNPs. TEM observations were carried out at the Center for the Development of Nanoscience and Nanotechnology (University of Santiago de Chile) using a HITACHI HT-7700 (Tokyo, Japan) instrument operated at an accelerating voltage of 120 kV with a Tungsten filament, allowing a resolution of 0.2 nm. Samples (powder) were dispersed in distilled water using an ultrasonic bath and deposited on a copper TEM grid (300 mesh) coated with Formvar/carbon. Water was subsequently evaporated at room temperature. The size of the SeNPs was determined via ImageJ software (version 1.53t) [43].

### 2.8. AFM

The morphology and surface topography of SeNPs were confirmed through AFM imaging. Samples were dispersed in water, and then 50 μL was added on a positively charged microscope slide (Porlab®, Nanjing, P.R China). The drop was left for 1 min for the nanoparticles (NPs) to adsorb. The glass was rinsed with plenty of deionized water and left to air dry for 15 min. The dried SeNPs were scanned by AFM (Witec Alpha 300) in AC mode at room temperature using an Hi’Res-C14/Cr-Au probe with a thickness of 2.1 μm, length of 125 μm, width of 25 μm, resonance frequency of 160 kHz, and a force constant of 5 N/m. To create the histogram, SeNPs were selected at random, and their size profile was determined. From this profile, the maximum was taken as the size of the nanoparticle.

### 2.9. Energy-Dispersive X-ray Spectroscopy (EDS)

The elemental composition of the purified NPs was determined with SEM coupled to EDS. SeNPs (powder) were carefully mounted on SEM stubs using adhesive tape uniformly coated with carbon (LEICA EM ACE200, Wetzlar, Germany) and then placed in a sample chamber of SEM-EDS (Zeiss EVOMA10, Oberkochen, Germany—Oxford Instruments X-act EDX system, Wycombe, UK). They were scanned at 1500× magnification and a voltage of 20 kV.

### 2.10. Raman Spectroscopy

Raman measurements were performed to explore the crystalline traits and lattice dynamics of the synthesized SeNPs. Solid SeNPs were placed on a clean cover slip. Normal Raman spectra were recorded using Witec Alpha 300 equipment with an excitation wavelength of 633 nm (spectral range 150–2100 cm^−1^). The acquisition interval was 1 s, and all spectra were averaged over 32 independent runs. The acquired digital experimental spectroscopic data were processed and plotted using Origin 8.5.

### 2.11. X-ray Diffraction

XRD patterns were performed using a RIGAKU Multipurpose diffractometer model Smartlab, with Theta-Theta Bragg-Brentano geometry goniometer and coupled to a solid-state detector D/teX Ultra 250 (Rigaku Corporation, Akishima-shi, Japan) for data acquisition. The X-ray beam was generated using a copper tube target (Cu Kalpha λ = 1.5418Å) at 30 kV/10 mA and filtered with Ni. The measure was recorded at room temperature between 5 and 55° 2-theta (degrees) with a step of 0.02° and a scan speed of 1°/min. Optical configurations were adjusted by divergent and receiving slits on both sides, with parallel Soller slits of 5° and 5 mm, respectively. The instrumental resolution was aligned with NIST SRM660c standard (LaB6). While the XRD measurement was carried out, the sample films were positioned on a silicon wafer with an orientation of <100> (about 69°). Subsequently, the patterns obtained were slightly smoothed using the Savitzky–Golay model, and any strange peaks coming from the silicon wafer were mathematically removed. The data obtained were optimized by TOPAS Academic V6 Program by modeling the peak shapes [44].

### 2.12. FTIR Spectroscopy

FTIR measurements were performed to identify the functional groups involved in the synthesis of biogenic SeNPs. The FTIR spectra of the dried samples were recorded on an ALPHA FTIR-ATR Bruke Spectrometer on a wavenumber range of 400–4000 cm^−1^ at a resolution of 1 cm^−1^ and 100 scans. The baseline was corrected for each spectrum using the “automatic baseline correct” function, and the spectrum was smoothed using the software’s standard “automatic smooth” function, which employs the Savitsky–Golay algorithm (95-point moving second-degree polynomial).

### 2.13. X-ray Photoelectron Spectroscopy

XPS measurements were performed using an X-ray photoelectron spectrometer (SPECS, Berlin, Germany). A FlexPS system from SPECS was equipped with a hemispherical analyzer PHOBIOS 150 and detector 1D-DLD, with a monochromatic X-ray source FOCUS 500 providing Ag K-alpha radiation with a characteristic energy of 2984 eV, using spot size of 500 μm and depth of 10 nm. Analyses were performed using two different points of powdered samples, and the elemental composition was analyzed using CasaXPS software (version 2.3.19, Casa Software Ltd, Las Vegas, NV, USA).

## 3. Results

### 3.1. Bacterial SeNP Synthesis

The biological synthesis of SeNPs was carried out using two endophytic selenobacteria (*Bacillus* sp. E5 and *Enterobacter* sp. EC5.2) previously selected for their ability to survive aerobically at high concentrations of selenite, as well as for their PGP traits. According to Durán et al. [13], *Bacillus* sp. E5 and *Enterobacter* sp. EC5.2 presented MIC values for selenite of 120 mM and 100 mM, respectively. Firstly, it was observed that both bacterial cultures turned red at all incubation times tested (Appendix A). The characteristic red color has been reported to be direct proof of the selenite reduction to zero-valent elemental selenium [45]. On the contrary, no color change was observed in the control experiment carried out with the selenite-containing medium without bacterial inoculation, thus indicating the active participation of these bacteria in selenite reduction. Moreover, once cell suspensions were centrifuged, it was possible to visualize that although both the pellet and the cell-free medium were reddish yellow, the red color was found to be more intense in the pellet than in the supernatant (Appendix A). This may suggest that both strains have the capacity for intracellular and extracellular SeNP production, maybe being more noticeable intracellularly.

Then, considering these observations, the total Se content accumulated by both endophytic strains grown in the presence of 5 mM sodium selenite (acting as a precursor of the bioreaction) was determined through HG-AAS. The results showed that both selenobacteria strains were able to accumulate high Se concentrations in comparison to the control without Se and that the red pellet of *Bacillus* sp. E5 presented a higher Se concentration than that of *Enterobacter* sp. EC5.2, with values of 12016 ± 4205 mg kg^−1^ and 3543 ± 1240 mg kg^−1^, respectively (Appendix A).

### 3.2. Localization of SeNPs in *Bacillus* sp. E5 and *Enterobacter* sp. EC5.2 Cultures

The ability of both bacterial strains to convert selenite to SeNPs was also visualized by microscopy. To this end, culture samples were collected after 6 and 24 h of growth in the absence and presence of selenite 5 mM. Then, the samples were processed and analyzed by TEM. Se nanospheres were observed after 6 h of incubation in both bacterial cultures supplemented with selenite (Figure 1). These NPs were not observed in the bacterial cells grown on Tris-HCl pH 8 without the oxyanion. Interestingly, unlike *Enterobacter* sp. EC5.2, the Gram-positive bacteria *Bacillus* sp. E5 clearly presented SeNPs and rod-like structures on the edge of the bacterial cell simultaneously (Figure 1(a2)).

The micrographs showed that the SeNPs were located predominantly in the extracellular environment in the surrounding medium or attached to the cell surface (Figure 1a,b). Moreover, these NPs seemed to be entrapped on components actively secreted by bacterial cells, such as extracellular polymeric substance (EPS) (Figure 1a,b).

AFM experiments were also performed to explore the nanoscale surface morphology of both endophytic strains producing various amounts of SeNPs. AFM topography and the amplitude images of *Bacillus* sp. E5 and *Enterobacter* sp. EC5.2 after treatment with 5 mM Na_2_SeO_3_ for 6 h are shown in Figure 2, together with AFM images of untreated bacterial cultures. Topography images provided information about differences in surface and structure, whereas amplitude images allowed better observation of surface features and fine details [46]. Based on AFM images, bacterial cultures without selenite presented the typical rod shape of *Bacillus* spp. and *Enterobacter* spp. (Figure 2(a1,a2,b1,b2)). After 6 h of incubation with 5 mM of Na_2_SeO_3_, SeNPs were observed in both bacterial cultures. These images corroborated the observations obtained by TEM in the sense that SeNPs were either attached to the cell wall or located in the surrounding medium, without relevant variations in their volume and shape (Figure 2(a3,a4,b3,b4)). AFM revealed thick and robust filaments in both cases distributed randomly around the cell surface, as well as the flattened structures surrounding the cells, in which the SeNPs were trapped. They represent collapsed cell envelopes generated by the drying process [47]. The cell wall of *Bacillus* sp. E5 presented more features than that of *Enterobacter* sp. EC5.2, which may be related to the fact that Gram-positive bacteria are surrounded by layers of peptidoglycan many times thicker than are found in Gram-negative bacteria.

### 3.3. Chemical and Physical Characteristics of Biogenic SeNPs

#### Stability, Morphology, and Size of SeNPs

Intracellular and extracellular SeNPs produced by *Bacillus* sp. E5 (named B-SeNPs) and *Enterobacter* sp. EC5.2 (named E-SeNPs) after 4, 6, 24, and 48 h exposure to 5 mM selenite were purified and characterized to observe the effect of incubation time on their stability and size. The hydrodynamic size, PDI, and zeta potential of biosynthesized SeNPs were measured with DLS. Most NPs produced intracellularly by both bacteria at different incubation times presented PDI values below 0.3 (Table 1a), indicating that samples were monodispersed [48]. Interestingly, B-SeNPs obtained after 24 and 48 h incubation presented PDI values (mean ± standard deviation) of 0.33 ± 0.05 and 0.42 ± 0.06, respectively. In addition, all B-SeNP formulations had an average zeta potential (mean ± standard deviation) between −31.2 ± 1.20 and −22.9 ± 1.32 mV, whereas E-SeNPs presented zeta potential values slightly higher than those of B-SeNPs (ranging from −35.5 ± 0.9 to −27.8 ± 1.79 mV), giving them better electrostatic stability [35]. With respect to hydrodynamic size, there was a general trend showing that the average diameter of SeNPs in both formulations increased with increasing incubation time. Remarkably, both formulations reached their maximum size with incubation times of 24–48 h. Similarly, the size of extracellular SeNPs directly depended on the incubation time (Table 1b). However, DLS analysis revealed that SeNPs found in the extracellular space were larger than intracellular NPs. Furthermore, extracellularly synthesized SeNPs had high PDI values (>0.3) in every case, corresponding to a larger size distribution and signs of aggregation.

Thus, based on the results presented in Table 1, the smallest SeNPs were intracellularly synthesized at 6 h of incubation; the average diameter of B-SeNPs was 112 ± 3 nm, while that of E-SeNPs was 119 ± 2 nm. Since the size of NPs is a key factor for their chemical and biological properties, these two SeNPs were the formulations selected for further analysis. The quantitative size analysis of biosynthesized SeNPs was implemented with TEM and AFM. Firstly, TEM images revealed that B-SeNPs and E-SeNPs had a spherical shape with homogeneous size distribution (Figure 3). Moreover, TEM images determined that the average size of B-SeNPs was 83.44 ± 2.90 nm, whereas E-SeNPs presented an average size of 56.23 ± 4.85 nm (Figure 3c,d). Therefore, TEM analysis confirmed the formation of SeNPs with a size below 100 nm—the smallest particles of those produced by the Gram-negative strain.

The morphology and surface topography of SeNPs were confirmed via AFM imaging. Figure 4a,b show the 2D topographical images of B-SeNPs and E-SeNPs deposited on the glass surface, respectively, while Figure 4c,d show the diameter histograms of the SeNPs. AFM results showed that the SeNPs were spherical in shape, with average sizes of 140.66 ± 2.71 nm for B-SeNPs and 97.31 ± 4.13 nm for E-SeNPs.

When comparing the diameters of E-SeNPs measured by DLS, TEM, and AFM, it becomes apparent that the size measured by DLS was slightly larger than the size recorded by the two microscopy techniques. However, it should be noted that the diameter value of B-SeNPs obtained by AFM is the highest compared to the ones obtained by DLS and TEM.

The elemental composition of SeNPs was evaluated by EDS. Since the biogenic NPs were dried on carbon wafers, the C signal (Kα = 0.277 KeV) was obtained for all samples analyzed (Appendix A). However, it should be noted that the C signal could also be derived from the coating of the NPs. The EDS spectrum corroborated the presence of Se in both NPs showing the specific Se peak (Lα = 1.379 KeV) and accounting for ≈ 9.50 % of the total component elements (Appendix A). Moreover, other marked peaks were also detected, such as nitrogen (N; Kα = 0.392 keV), oxygen (O; Kα = 0.525 KeV), phosphorus (P; Kα = 2.013 KeV), and sulfur (S; Kα = 2.307 KeV). This could indicate that the material covering the SeNP surface was organic in nature.

### 3.4. Crystallographic Structure of SeNPs

The crystallographic structure of the SeNPs was analyzed by XRD. Based on the results shown in Figure 5, the XRD pattern of both SeNPs presented wide, overlapping curves with no sharp Bragg reflections, indicating that the red SeNPs generated by bacterial reduction are amorphous in structure [49]. The Raman spectroscopy analysis of the biosynthesized SeNPs is shown in Figure 6. The spectrum of both NPs exhibited a very strong band with a maximum at 254 cm^−1^, attributed to the A1 stretching mode of amorphous Se. Moreover, a weak but noticeable band at 500 cm^−1^ was also observed in all the SeNPs, corresponding to homonuclear stretching S–S bonds.

### 3.5. Biomolecules Bound to Biosynthesized SeNPs

The functional groups and biomolecules surrounding SeNPs were identified by FTIR spectroscopy analysis on bacterial cells and purified NPs. The spectrum exhibited in Figure 7 indicates the existence of biological macromolecules, such as lipids, proteins, and carbohydrates, likely acting as capping agents and most probably contributing to SeNP stability. The major absorbance bands observed in the nanoparticle and bacterial cell spectra (between 4000 and 400 cm^−1^) were at 3280, 2927, 1625, 1592, 1450, 1390, 1233, and 1054 cm^−1^ (Appendix A). As shown in Figure 7a–c, the main characteristic peaks found in the FTIR spectrum of SeNPs were the same as those of bacterial cells in both cases, thus indicating that selenium nanoparticles are surrounded by active substances of bacterial cells. It was also observed that B-SeNPs possessed absorption peaks with higher intensity than those obtained for E-SeNPs (Figure 7c), which means an increase in the amount of the functional groups on the SeNPs produced by the Gram-positive strain *Bacillus* sp. E5. X-ray photoelectron spectroscopy (XPS) analysis was also performed to expose the bonding mechanism and element valence of SeNPs. Signals corresponding to C 1s, N 1s, and O 1s were detected (Figure 8). In addition, the peak found at 55.38 represents the 3d peak of typical Se, this signal being much weaker than that of C 1s, N 1s, and O 1s.

## 4. Discussion

In recent years, the use of SeNPs in agriculture has been widely described as a potential strategy to stimulate plant growth, biofortify crops, and control plant disease [31,32,50]. In this context, the present study strengthens the possibility of using two contrasting endophytic isolates as eco-friendly “cell factories” for producing SeNPs with different structural and physico-chemical properties for further use as biofortifying agents and/or other biotechnological purposes. Gram-positive and Gram-negative bacteria with high Se tolerance and plant-growth-promoting (PGP) attributes were used to produce SeNPs [13]. Considering that *Bacillus* sp. E5 was isolated from the tissue of wheat plants treated with Se, its Se tolerance was slightly greater (120 mM) than that observed in *Enterobacter* sp. EC5.2 (100 mM), which was isolated from the tissue of wheat plants not treated with Se [13]. However, the two bacterial strains showed high Se tolerance, similar to that reported in *Pseudomonas moraviensis* (120 mM) [51] and *Vibrio natriegens* (100 mM) [52].

Interestingly, Zhang et al. [53] reported that bacterial strains isolated from Se-rich habitats were usually quicker and more efficient than those from Se-free environments in reducing the Se oxyanion. These results are in accordance with our results, indicating that *Bacillus* sp. E5 accumulated the highest Se content in comparison to *Enterobacter* sp. EC5.2. This may suggest that the ability to incorporate the inorganic Se source into the bacterial metabolism and reduce it into other Se forms differs between the bacterial strains. It has been observed that different organic Se species, as well as SeNPs, are produced by bacteria grown on a Se-enriched medium [23]. Particularly, in cases of selenite exposure, selenomethionine (SeMet) is the main organic Se form found in bacterial inocula, whereas Se particles are the predominant Se species [14]. Interestingly, different types of bacteria produce different morphologies of Se nanostructures, and furthermore, the synthesized NPs possess different biomolecular compositions on their surface, which act as a cover or capping layer [54,55]. It has been pointed out by many authors that the biological composition of the outer organic layer plays a major role in the low toxicity and high biocompatibility of the NPs produced by bacteria [23,56,57].

Taking the above into consideration, we evaluated the bioproduction of SeNPs on the selected bacteria under optimal synthesis conditions.

The fast reduction of selenite (the appearance of the red color typical of colloidal Se) by both endophytic strains (after 6 h of incubation) could be attributed to the huge reducing power featuring metabolically active resting bacterial cells per given time [56,58], which is the kind of biotic transformation of selenite that was carried out in the present study. In fact, Presentato et al. [57,59] discovered that *Rhodococcus aetherivorans* BCP1 resting cells had a greater performance to produce TeNRs as compared to actively growing cultures.

The appearance of red color on bacterial cultures grown in the presence of selenite was accompanied by a characteristic garlic-like odor, suggesting that a volatilization step to methylated Se products occurs during the reduction process [53,58,60].

The presence of SeNPs in both intra- and extracellular compartments was observed by TEM and AFM (Figure 1 and Figure 2), indicating that the selected strains, *Bacillus* sp. E5 and *Enterobacter* sp. EC5.2, were able to transform selenite to SeNPs. Curiously, it was visualized that *Bacillus* sp. E5 simultaneously synthesized SeNPs and a few Se nanorods (SeNRs) on the cellular surface (Figure 1). In this regard, Piacenza et al. [48] extensively reported that the different bacterial physiological states determine morphological changes in Se nanomaterials, resulting in the synthesis of both NPs and NRs.

AFM images (Figure 2) revealed no evidence of cell lysis nor a distribution of SeNPs within the surface layers. At this point, it could be hypothesized that the cell wall would play an important role in the intracellular and extracellular mechanisms of the synthesis and transport of SeNPs [61,62,63]. Additionally, the adhesion of SeNPs observed to the cell surface could be explained with the extended Derjaguin–Landau–Verwey–Overbeek (DLVO) theory, based on the role of electrostatic double-layer interactions in adhesion, as observed in *Pseudomonas putida* strains by Hwang et al. [64].

However, the key factors involved in the reduction of selenite in the two endophytic strains studied here have not been elucidated, and the mechanism responsible for NP transport from the cell to the exterior deserves more investigation.

The purification and subsequent analysis by DLS (Table 1a,b) of intracellular and extracellular SeNPs demonstrated that the intracellular ones have smaller sizes than those found in the extracellular space, since in the external compartment, the NPs tend to agglomerate and lose their shape, increasing in size. Similar results were reported by Torres et al. [65] for *Pantoea agglomerans*, obtaining larger SeNPs in the extracellular space. Thus, the purification involving bacterial lysis allows for obtaining the smaller nanoparticles found in the intracellular space. Then, focusing only on the intracellular SeNPs, the results obtained by DLS (Table 1a,b) showed that PDI values increased slightly in both types of SeNPs (B-SeNPs and E-SeNPs), as the incubation period increased, showing a greater hydrodynamic diameter after 24–48 h than 4–6 h of incubation. These findings are in line with previous studies that show a similar trend with increased PDI and hydrodynamic size values during prolonged reaction time [66,67]. The negative zeta potential values differed moderately between the two types of NPs, with E-SeNP values being the highest in all cases; they would therefore be the more physically stable type. Given that the natural stability of biogenic NPs is conferred by the organic layer [63], the overall higher hydrodynamic diameter and lower zeta potential of B-SeNPs could indicate a lower stabilizing effect of Gram-positive compared to Gram-negative capping. Bulgarini et al. [55] also observed that SeNPs produced by Gram-positive bacteria had lower zeta potential values than SeNPs produced by Gram-negative bacteria. This is probably linked to the total amount of material covering the SeNP structure [40] or the composition and nature of molecules associated with these NPs [55]. The average size of SeNPs produced by *Bacillus* sp. E5 (~83 nm) and *Enterobacter* sp. EC5.2 (~56 nm), determined from TEM images (Figure 3), was consistent with the size of SeNPs synthesized by other microorganisms ranging from ~50 nm [66,68] to ~100 nm [54]. The SeNPs obtained from both strains are considered to be within the NP range (1–100 nm) [23] with potential applications in physical, chemical, and biological sciences, since they are more biocompatible with living systems and more beneficial than bigger structures (100–500 nm) [69]. The results recovered from TEM showed mainly the diameter of particle cores [70]; this explains the considerable discrepancies observed in particle sizes between TEM (size ~83 nm; ~56 nm) (Figure 3), AFM (size ~140 nm; ~97 nm) (Figure 4), and DLS (size ~112 nm; ~119 nm) (Table 1). This could be explained by that the latter two techniques might consider the organic layer surrounding the SeNPs as well as presenting a greater aggregation of particles in the samples. Moreover, a higher diameter value of B-SeNPs obtained by AFM could be justified mainly because B-SeNPs may tend to form aggregates or be covered with abundant organic material, which provokes an effect of the cantilever tip–particle interaction during imaging, resulting in the imaged particles having higher diameter values than the intact particles [71].

The EDS spectra of the spherical nanoparticles confirmed the presence of Se by the observation of the Se-specific absorption peak at 1.379 KeV in both cases (Appendix A) [72]. Furthermore, signals for N, O, P, and S were also detected, being related to protein molecules or even associated with the presence of other biomolecules (e.g., phospholipids, nucleic acids) [73]. Interestingly, a higher percentage of S was found surrounding the structure of E-SeNPs than that found in B-SeNPs, suggesting that sulfur plays an important role in the formation of E-SeNPs [74]. This result was also corroborated by the Raman spectrum of the purified SeNPs (Figure 6), in which, aside from noting the characteristic amorphous Se band at ~254 cm^−1^, the typical S–S stretching band (~500 cm^−1^) was observed [75] to be more prominent in E-SeNPs. This, again, could be correlated with a substantial amount of sulfur in the SeNPs, especially in E-SeNPs. In fact, several studies [76,77] have shown that the constant presence of an S signal may be due to the involvement of RSHs in SeO_3_^2−^ bioprocessing for bacterial cultures. Moreover, the XRD data also confirmed that both SeNPs are amorphous in structure (Figure 5). This result agrees with the fact that most SeNPs produced by biological reduction are typically amorphous in nature [78,79,80]. The amorphous nature of biogenic SeNPs is an intrinsic feature of the redox reaction for their production. Indeed, SeNPs obtained from reduction or redox processes are always amorphous in nature [35].

According to the FTIR spectroscopic data (Figure 7), the most prominent biomacromolecular composition of the two SeNPs obtained includes proteins (for which the amide I and II bands are accompanied by the weaker absorption around 1230 cm^−1^ featuring another typical protein-related band, amide III) [75,81], polysaccharides (see the characteristic carbohydrate structure at 1233 cm^−1^), and lipids with less intense bands (featured via a combination of stretching and binding C–H vibrations with the CH3 of acyl chains) [80]. Lastly, traces of water (which can form strong H-bonds with polar biomolecular groups and thus might be not fully removed by freeze-drying; see the Materials and Methods section) could contribute to the increased stretching of the O–H region (3600–3200 cm^−1^). As Wu et al. [81] have already reported, FTIR also corroborated that BioSeNPs inherited the active substances from bacterial cells, suggesting the important role of these biomolecules surrounding SeNPs in the synthesis and stabilization [56,57]. The higher absorption peaks observed in B-SeNPs compared to those obtained for E-SeNPs could be related to the fact that Gram-positive bacteria possessed many components actively secreted by bacterial cells, as well as a thick peptidoglycan layer (Figure 1 and Figure 2). Finally, XPS data were obtained on the bonding mechanism and element valence of SeNPs (Figure 8), which are consistent with the EDS analysis. However, in the case of E-SeNPs, N was detected in the XPS spectrum but not in the EDS spectrum. This could be due to the overlap of C and O on the N peak in the EDS spectrum [82]. Additionally, within the XPS data, it is noteworthy that the Se 3d signal is much weaker than that of C 1s, O 1s, and N 1s. This is because XPS is a surface analysis method. Even though X-rays can penetrate SeNPs, only the photoelectrons emitted by C, O, and N in the shell layer on the surface of the sample can escape freely, while the photoelectrons emitted by Se inside the nanoparticle can do so only with difficulty, generating a weak Se 3d signal [82]. Similar results have already been published by previous studies [40,81].

## 5. Conclusions

Here, we report for the first time that two contrasting endophytic selenobacteria, *Bacillus* sp. E5 (*Bacillus paranthracis*) and *Enterobacter* sp. EC5.2 (*Enterobacter ludwigi*), under optimal synthesis conditions (incubation time, the concentration of oxyanion, and physiological state) can produce intra- and extracellular SeNPs. According to the size, intracellular SeNPs were chosen to analyze their structural and physico-chemical properties more deeply. Briefly, B-SeNPs have a larger size, have a lower Z-potential, and possess a more robust capping layer compared to E-SeNPs. The present findings indicate that E-SeNPs are the smallest formulations, whereas B-SeNPs are enveloped with the hugest organic layer. Thus, considering that the size and the biomolecular composition of the capping layer of biogenic NPs are key factors for their stability and bioactivity, our further efforts will be channeled into the evaluation of how the different traits of individual SeNPs modulate their biological actions.

## Figures and Tables

**Figure 1 microorganisms-11-01600-f001:**
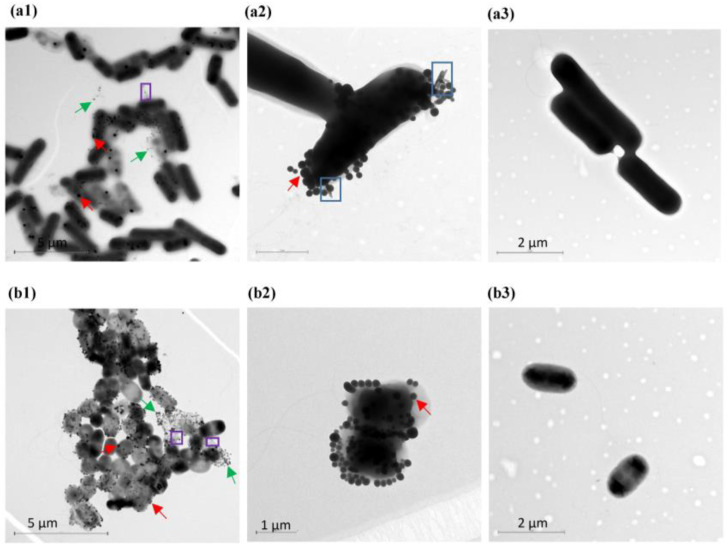
TEM images of cells of *Bacillus* sp. E5 (**a1**,**a2**) and *Enterobacter* sp. EC5.2 (**b1**,**b2**) after 6 h of growth in the presence of 5 mM Na_2_SeO_3_. Red arrows indicate SeNPs located on the cell surface, whereas green arrows indicate SeNPs located in the surrounding medium. Purple frames show components actively secreted by bacterial cells. Blue frames on image (**a2**) show nanorods (SeNRs). Images (**a3**,**b3**) show bacterial cells grown without selenite.

**Figure 2 microorganisms-11-01600-f002:**
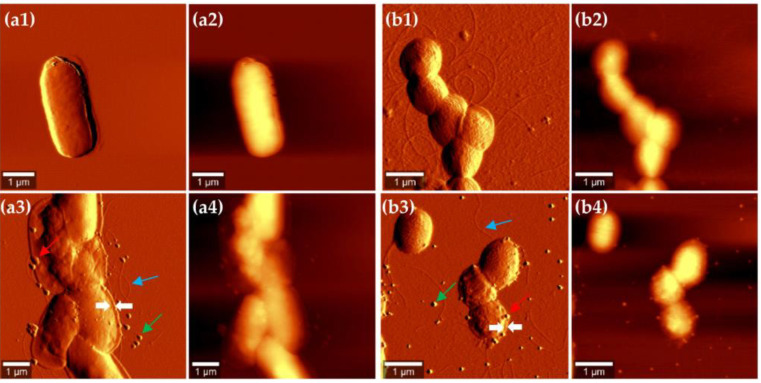
AFM images of *Bacillus* sp. E5 (**a**) and *Enterobacter* sp. EC5.2 (**b**) after 6 h of growth in the absence (**a1**,**a2**,**b1**,**b2**) or in the presence (**a3**,**a4**,**b3**,**b4**) of 5 mM Na_2_SeO_3_. The first column of each bacterial strain shows amplitude images (**a1**,**a3**,**b1**,**b3**), whereas topography images are shown in the second column (**a2**,**a4**,**b2**,**b4**). Red arrows indicate SeNPs attached to the cell wall, whereas green arrows indicate SeNPs located in the surrounding medium. The white arrows correspond to the collapsed cell envelope (see details in main text). The blue arrows correspond to bacterial filaments.

**Figure 3 microorganisms-11-01600-f003:**
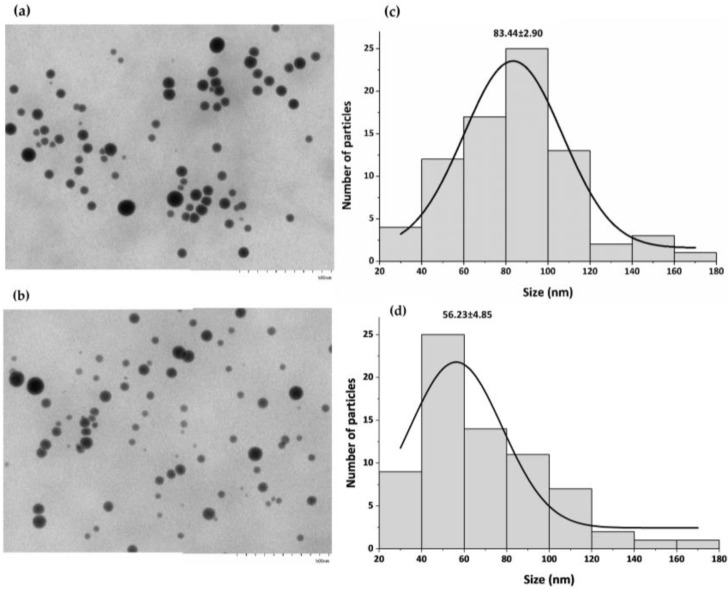
TEM observation of the purified SeNPs produced by *Bacillus* sp. E5 (**a**) and *Enterobacter* sp. EC5.2 (**b**) showing their spherical shape. The size distribution of SeNPs produced by *Bacillus* sp. E5 (**c**) and *Enterobacter* sp. EC5.2 (**d**).

**Figure 4 microorganisms-11-01600-f004:**
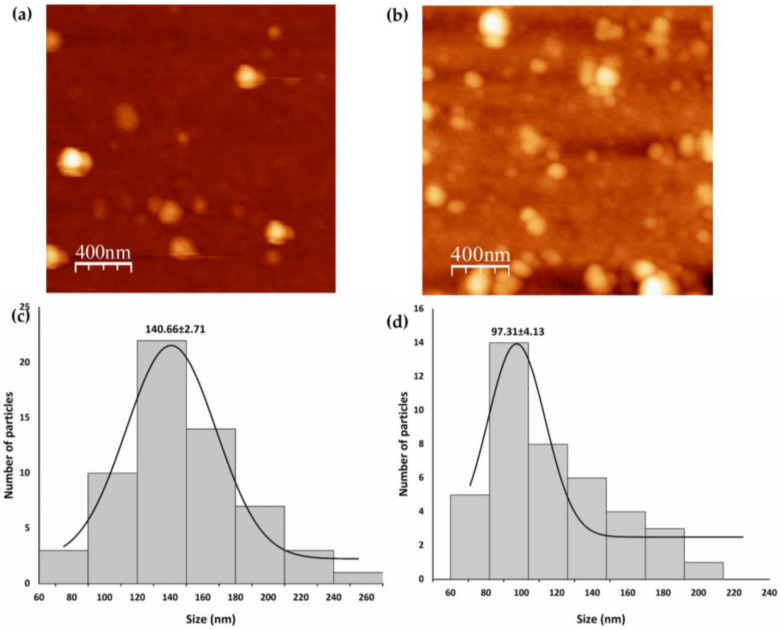
Characterization of SeNPs synthesized by *Bacillus* sp. E5 (B-SeNPs) and *Enterobacter* sp. EC5.2 (E-SeNPs). Two-dimensional AFM topographical image of B-SeNPs (**a**) and E-SeNPs (**b**). The graphics below (**c**,**d**) show the AFM histogram of B-SeNP and E-SeNP diameters.

**Figure 5 microorganisms-11-01600-f005:**
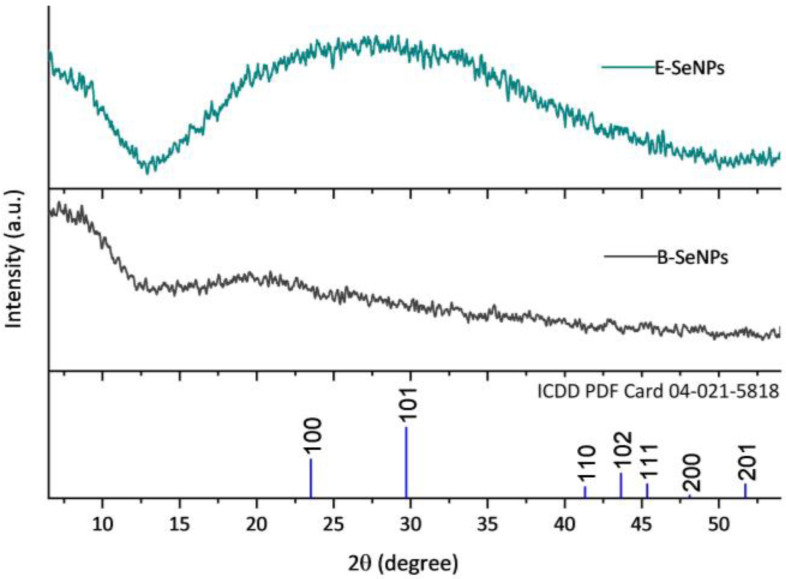
XRD pattern of SeNPs synthesized using *Bacillus* sp. E5 (B-SeNPs) and *Enterobacter* sp. EC5.2 (E-SeNPs).

**Figure 6 microorganisms-11-01600-f006:**
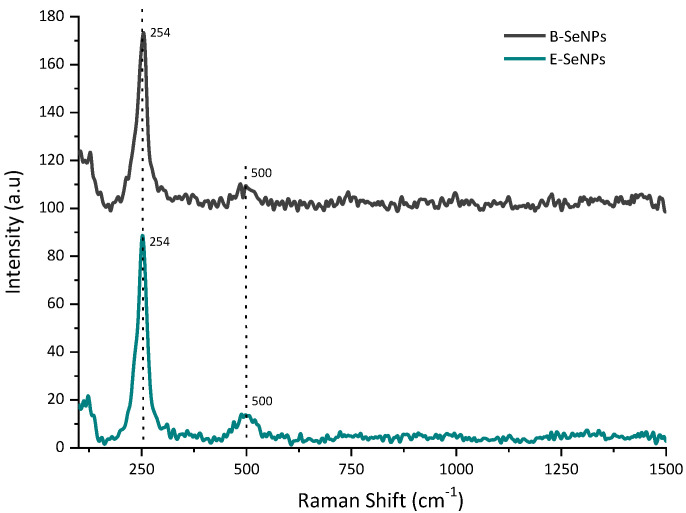
Raman spectrum at 532 nm excitation wavelength of SeNPs synthesized by *Bacillus* sp. E5 (B-SeNPs) and *Enterobacter* sp. EC5.2 (E-SeNPs) and incubated with 5 mM of sodium selenite within 6 h. The signal of Se–Se vibrations was observed at 254 cm^−1^.

**Figure 7 microorganisms-11-01600-f007:**
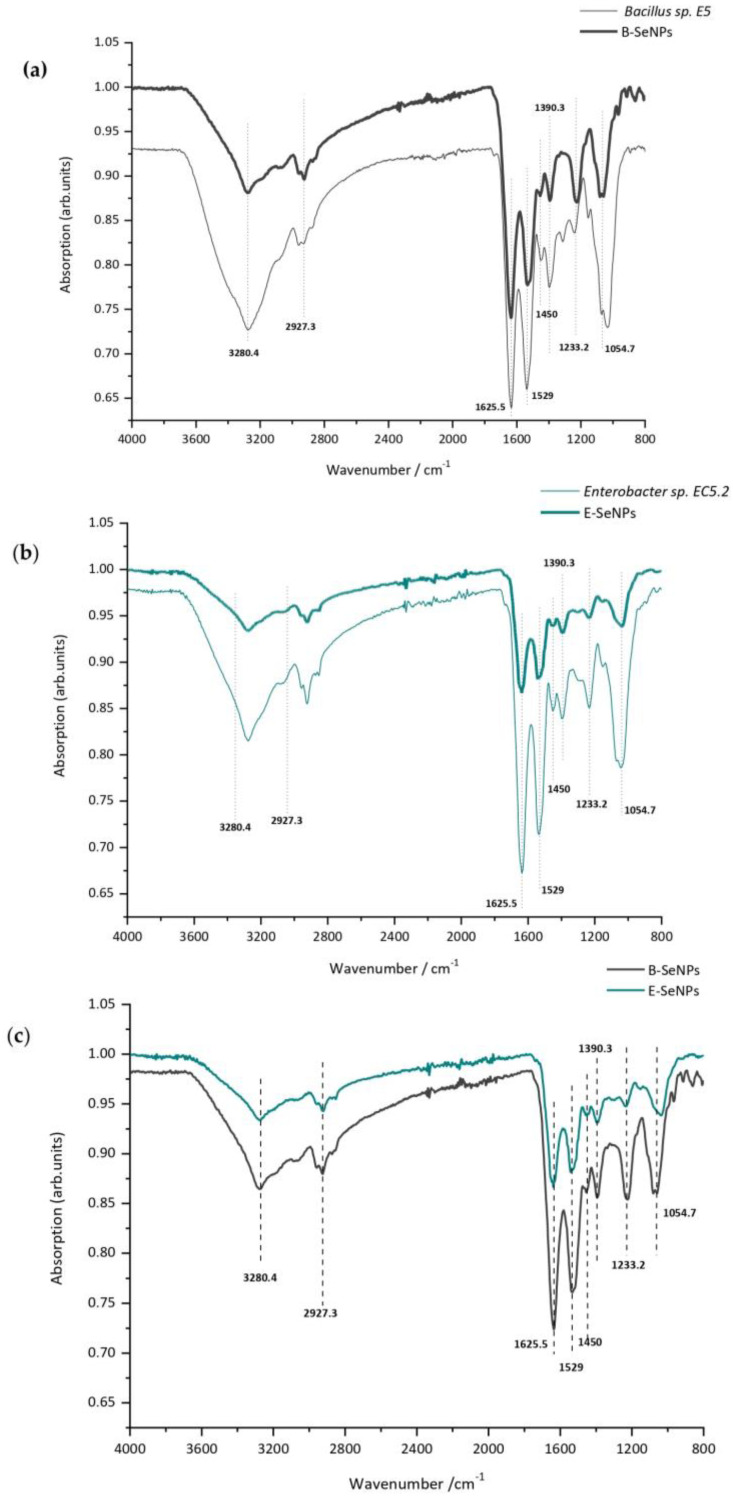
FTIR spectrum of bacterial cells compared to the one obtained from purified SeNPs (**a**,**b**) and the FTIR spectrum of both SeNPs synthesized by *Bacillus* sp. E5 (B-SeNPs) and *Enterobacter* sp. EC5.2 (E-SeNPs) (**c**).

**Figure 8 microorganisms-11-01600-f008:**
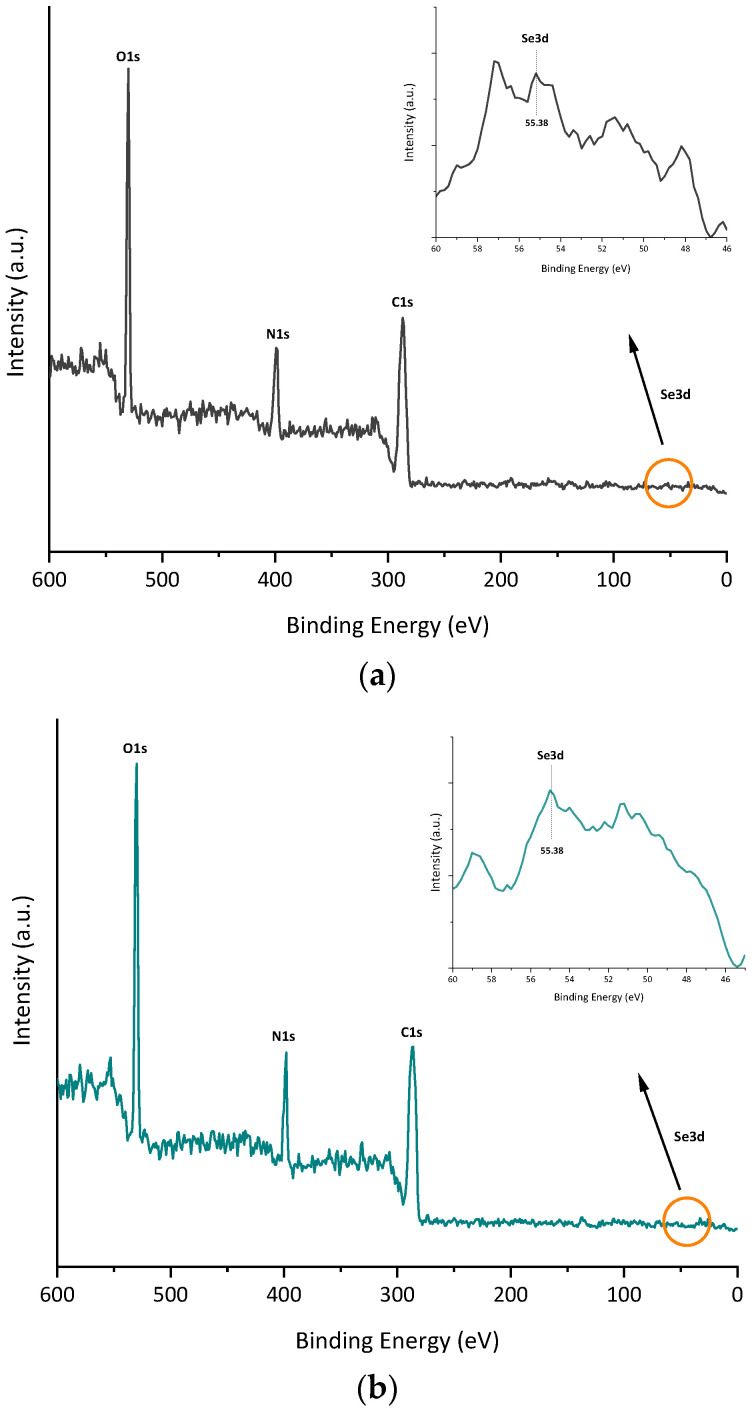
XPS spectra: (**a**) B-SeNPs and (**b**) E-SeNPs.

**Table 1 microorganisms-11-01600-t001:** Particle size, PDI, and the zeta potential of intracellular (a) and extracellular (b) SeNPs biosynthesized by *Bacillus* sp. E5 (B-SeNPs) and *Enterobacter* sp. EC5.2 (E-SeNPs) at different incubation times with 5 mM of sodium selenite (Na_2_SeO_3_) as the sole Se source.

**(a)**			
**Nanoparticles**	**Particle Size** **(nm)**	**Polydispersity Index (PDI)**	**Zeta Potential** **(mV)**
B-SeNPs—4 h E-SeNPs—4 h	114 ± 1 133 ± 12	0.28 ± 0.02 0.25 ± 0.03	−31.2 ± 1.20 −35.16 ± 0.54
B-SeNPs—6 h E-SeNPs—6 h	112 ± 3 119 ± 2	0.20 ± 0.02 0.23 ± 0.01	−24.3 ± 0.62 −27.8 ± 1.79
B-SeNPs—24 h E-SeNPs—24 h	198 ± 11 143 ± 21	0.33 ± 0.05 0.30 ± 0.03	−25.7 ± 0.09 −28.42 ± 0.83
B-SeNPs—48 h E-SeNPs—48 h	195 ± 7 142 ± 2	0.42 ± 0.06 0.28 ± 0.02	−22.9 ± 1.32 −35.5 ± 0.9
**(b)**			
**Nanoparticles**	**Particle Size** **(nm)**	**Polydispersity Index (PDI)**	**Zeta Potential** **(mV)**
B-SeNPs—4 h E-SeNPs—4 h	216 ± 16 212 ± 13	0.34 ± 0.03 0.31 ± 0.04	−33.13 ± 4.90 −28.9± 0.54
B-SeNPs—6 h E-SeNPs—6 h	267 ± 34 216 ± 30	0.35 ± 0.01 0.35 ± 0.03	−27.8 ± 5.45 −23.5 ± 1.79
B-SeNPs—24 h E-SeNPs—24 h	280 ± 17 255 ± 32	0.54 ± 0.14 0.38 ± 0.04	−35 ± 2.52 −34.4 ± 0.83
B-SeNPs—48 h E-SeNPs—48 h	288 ± 18 276 ± 26	0.51 ± 0.04 0.41 ± 0.05	−33.4 ± 7.77 −31.8 ± 0.9

Data are presented as mean ± standard deviation.

## Data Availability

Not applicable.

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
