# Peer review of "A Comparative Study of the Synthesis and Characterization of Biogenic Selenium Nanoparticles by Two Contrasting Endophytic Selenobacteria"

_microorganisms, 2023, doi:10.3390/microorganisms11061600_

Round 1

Reviewer 1 Report

Recommendation letter

Dear Dr.

I have carefully read the MS # microorganisms-2399419-peer-review-v1 will considered for publication in microorganisms  under title of “A comparative study of the synthesis and characterization of biogenic selenium nanoparticles by two contrasting endophytic selenobacteria”, and I have the following concerns about the current form of manuscript.

1.  the manuscript is very useful for researchers in this field, but it is need a lot of revisions and rewrite based on standard English language especial in parts of abstract, materials and methods and results and discussion before provide the final form of publishing.

2.  Standard written as structures, spacing, units …..etc. should be revised, for example at line 122, sp. E5 it is not italic. no should be corrected. sp. EC5.2 it is not italic.……… etc.

3.  Accession no , must be corrected.

4.  Agar, not capital, it is not scientific name.

5.  In part of M&M, you should write clear experiments. From line 196 to 201, please, revise it.

6.  In results part, found standard roles for written must be done.

7.    Some cites should be added to discuss the study like: https://doi.org/10.1016/j.bcab.2019.101080, https://doi.org/10.33263/BRIAC104.834842, https://doi.org/10.1016/j.ijbiomac.2020.08.063, https://doi.org/10.33263/BRIAC135.419.

8.    The name of bacteria is mentioned in abstract but the identification part was lacked .

9.  The toxicity evaluation should be mentioned.

10.          The English language needs to improve.

Recommendation letter

Dear Dr.

I have carefully read the MS # microorganisms-2399419-peer-review-v1 will considered for publication in microorganisms  under title of “A comparative study of the synthesis and characterization of biogenic selenium nanoparticles by two contrasting endophytic selenobacteria”, and I have the following concerns about the current form of manuscript.

1.  the manuscript is very useful for researchers in this field, but it is need a lot of revisions and rewrite based on standard English language especial in parts of abstract, materials and methods and results and discussion before provide the final form of publishing.

2.  Standard written as structures, spacing, units …..etc. should be revised, for example at line 122, sp. E5 it is not italic. no should be corrected. sp. EC5.2 it is not italic.……… etc.

3.  Accession no , must be corrected.

4.  Agar, not capital, it is not scientific name.

5.  In part of M&M, you should write clear experiments. From line 196 to 201, please, revise it.

6.  In results part, found standard roles for written must be done.

7.    Some cites should be added to discuss the study like: https://doi.org/10.1016/j.bcab.2019.101080, https://doi.org/10.33263/BRIAC104.834842, https://doi.org/10.1016/j.ijbiomac.2020.08.063, https://doi.org/10.33263/BRIAC135.419.

8.    The name of bacteria is mentioned in abstract but the identification part was lacked .

9.  The toxicity evaluation should be mentioned.

10.          The English language needs to improve.

Author Response

Reviewer #1,

I have carefully read the MS # microorganisms-2399419-peer-review-v1 will considered for publication in microorganisms under title of “A comparative study of the synthesis and characterization of biogenic selenium nanoparticles by two contrasting endophytic selenobacteria”, and I have the following concerns about the current form of manuscript.

Response: We thank the reviewer for his/her thoughtful and thorough review and believe his/her input has been invaluable to make our manuscript of higher quality.

  • The manuscript is very useful for researchers in this field, but it is need a lot of revisions and rewrite based on standard English language especial in parts of abstract, materials and methods and results and discussion before provide the final form of publishing.

Response:  We sincerely appreciate this comment. We have polished this manuscript by MDPI Language Editing Services (the official certificate is attached).

  • Standard written as structures, spacing, units …..etc. should be revised, for example at line 122, sp. E5 it is not italic. no should be corrected. sp. EC5.2 it is not italic.……… etc.

Response:  We have changed it into the correct style. The corrections are highlighted in the manuscript by track changes.

  • Accession no , must be corrected.

Response:  We have changed it into the correct style (lines 132-133).

  • Agar, not capital, it is not scientific name.

Response:  We have changed it into the correct form (line 136).

  • In part of M&M, you should write clear experiments. From line 196 to 201, please, revise it.

Response:  We thank the reviewer for this valuable comment. This part of the methodology has been improved in a clearer way (lines 191-203).

  • In results part, found standard roles for written must be done.

Response:  We have changed it into the correct style. The corrections are highlighted in the manuscript by track changes.

  • Some cites should be added to discuss the study like: https://doi.org/10.1016/j.bcab.2019.101080, https://doi.org/10.33263/BRIAC104.834842, https://doi.org/10.1016/j.ijbiomac.2020.08.063, https://doi.org/10.33263/BRIAC135.419.

Response:  Two of the suggested references have been added to the manuscript. The first one (https://doi.org/10.1016/j.bcab.2019.101080) corresponds to the reference nº 31 (line 90 – introduction section) and the second one (https://doi.org/10.33263/BRIAC104.834842) corresponds to the reference nº 78 (line 781 – discussion section).

  • The name of bacteria is mentioned in abstract but the identification part was lacked .

Response:  It has been corrected (lines 24-26).

  • The toxicity evaluation should be mentioned.

Response:  We have mentioned the role of biogenic capping agents in the toxicity and biocompatibility of SeNPs produced biologically (lines 87-88 – introduction section; lines 697-699 – discussion section).

  • The English language needs to improve.

Response:  We sincerely appreciate this comment. We have polished this manuscript by MDPI Language Editing Services.

Reviewer 2 Report

This is an interesting research work. However, some issues need further clarification by the authors to deepen the understanding of the results of this study.

1. Lines 117-119: The expression here is unclear and overly broad. Further clarification on the significance and potential impact of the research in this paper is needed by the authors. It is suggested that the implications of this study can be further revealed in further discussions.

2. Figures 1 and 2: For better comparison, images of the control group are needed from the authors.

3. In the discussion section, this study needs to further explore the interaction between particles and cells and the possible effects of this interaction on particle stability in the context of DLVO theory.

4. The conclusion section still reinforces the significance of this study.

Author Response

Reviewer #2,

This is an interesting research work. However, some issues need further clarification by the authors to deepen the understanding of the results of this study.

Response: We would like to thank the reviewer for taking the necessary time and effort to revise this manuscript. We sincerely appreciate all your valuable comments and suggestions, which helped us in improving the quality of the manuscript.

  • Lines 117-119: The expression here is unclear and overly broad. Further clarification on the significance and potential impact of the research in this paper is needed by the authors. It is suggested that the implications of this study can be further revealed in further discussions.

Response: We have improved the expression related to the implications of this study, emphasizing the significance and potential impact of the present research. It has been clarified in lines 119-123 (introduction section) and 675-678 (discussion section).

  • Figures 1 and 2: For better comparison, images of the control group are needed from the authors.

Response: We have added the control images with their corresponding letter and numbers specified in captions and legends. In the case of Fig.1, the third image of each row shows bacterial cells grown without selenite (a3, b3). In the case of Fig.2, the AFM images “a1, a2, b1, b2” show bacterial cells grown without selenite.

  • In the discussion section, this study needs to further explore the interaction between particles and cells and the possible effects of this interaction on particle stability in the context of DLVO theory.

Response: We have made your recommended changes. The exerpt below has been included in the manuscript within the discussion section (lines 722-725):

Additionally, the adhesion of SeNPs observed to the cell surface could be explained with the extended Derjaguin–Landau–Verwey–Overbeek (DLVO) theory, based on the role of electrostatic double-layer interactions in adhesion, as observed in Pseudomonas putida strains by Hwang et al. [64].

  • The conclusion section still reinforces the significance of this study.

Response: Thank you for your valuable feedback.

Reviewer 3 Report

Minor comments

1)Lines 364-366: ‘the SeNPs were spherical in shape, with average sizes of 140.66 ± 2.71 for B-SeNPs and 97.31 ± 4.13 for E-SeNPs’- add units (nm) after numbers

2)Figure 2 ‘Figure 2. AFM images of Bacillus sp. E5 (A) and Enterobacter sp. EC5.2 (B) after 6 h of growth in the absence (A1, A2, B1, B2) or in the presence (A3, A4, B3, B4)’- there are no A, B indications on the pictures

3)Table 1- add statistics to the data- indicate the significance of differences between the values

4)Table 3 – change ‘A,B,C,D’ to ‘a,b,c,d’

5)Results section While describing Figures 3 and 4 indicate by a single phrase that while Figure 3 provides the size of pure nanoparticles, Figure 4 describes the size of coated nanoparticles. Otherwise, the reader ought to find the answer much later

6)Line 398- as far as I know, selenium nanoparticles are always amorphous and never crystalline. The presented proof is fine but it seems there is no necessity to give a profound discussion of the chance of the crystalline form presence.

recommendations: taking into account the significance of the results and the deep character of investigation I should recommend to submit your future work on the comparison of Se nanoparticles biological activity either in plants or/ and animals to the same Journal (Microorganisms) which will be both valuable for the Journal and important for the readers.

Author Response

Reviewer #3,

  • Lines 364-366: ‘the SeNPs were spherical in shape, with average sizes of 140.66 ± 2.71 for B-SeNPs and 97.31 ± 4.13 for E-SeNPs’- add units (nm) after numbers

Response: We have added the units (nm) after numbers. The corrections are highlighted in the manuscript by track changes.

  • Figure 2 ‘Figure 2. AFM images of Bacillus sp. E5 (A) and Enterobacter sp. EC5.2 (B) after 6 h of growth in the absence (A1, A2, B1, B2) or in the presence (A3, A4, B3, B4)’- there are no A, B indications on the pictures

Response: Sorry for this mistake. The corresponding indications have been added to the AFM images both in the legend and in the caption.

  • Table 1- add statistics to the data- indicate the significance of differences between the values

Response:  We appreciate your comment. However, at this point, we consider unnecessary to do statistics with the data presented in Table 1 for the following reason: the focus of this results is on selecting the synthesis time (according to the size and stability), instead of comparing the synthesis time of both bacteria. All measurements are independently. Therefore, we consider that it is not possible to do statistics from this data. We have found some works that follow our same line of thinking:

  1. Gallardo-Benavente, C., et al. Biosynthesis of CdS Quantum Dots Mediated by Volatile Sulfur Compounds Released by Antarctic Pseudomonas fragi. Front Microbiol 2019, 10,1866.
  2. Nile, S.H., et al. Antifungal Properties of Biogenic Selenium Nanoparticles Functionalized with Nystatin for the Inhibition of Candida albicans Biofilm Formation. Molecules 202328(4), 1836; https://doi.org/10.3390/molecules28041836.
  3. Fernández-Llamosas, H., Castro, L., Blázquez, M.L., Díaz, E., Carmona, M. Speeding up bioproduction of selenium nanoparticles by using Vibrio natriegens as microbial factory. Sci. Rep 2017, 7, 16046.
  4. Piacenza E, Presentato A,Ambrosi E, Speghini A, Turner RJ, Vallini G and Lampis S (2018). Physical–Chemical Properties of Biogenic Selenium Nanostructures Produced by Stenotrophomonas maltophilia SeITE02 and Ochrobactrum sp. MPV1. Front. Microbiol. 9:3178. doi: 10.3389/fmicb.2018.03178
  • Table 3 – change ‘A,B,C,D’ to ‘a,b,c,d’

Response: We have assumed that you could refer to Fig. 3 since Table 3 does not exist. However, following your suggestions, we have changed all capital letters (A,B,C,D) to small letters (a,b,c,d) in all figures and tables. The corrections are highlighted in the manuscript by track changes.

  • Results section While describing Figures 3 and 4 indicate by a single phrase that while Figure 3 provides the size of pure nanoparticles, Figure 4 describes the size of coated nanoparticles. Otherwise, the reader ought to find the answer much later

Response: Thank you for your clarification. The justification of the different sizes of SeNPs registered by TEM, AFM and DLS is explained in detail in lines 755-765 (discussion section).

  • Line 398- as far as I know, selenium nanoparticles are always amorphous and never crystalline. The presented proof is fine but it seems there is no necessity to give a profound discussion of the chance of the crystalline form presence.

Response: We totally agree with your remarks. In fact, SeNPs obtained from reduction or redox processes are always amorphous in nature (Piacenza, E., Presentato, A., Turner, R.J. Stability of biogenic metal(loid) nanomaterials related to the colloidal stabilization theory of chemical nanostructures. Critical Reviews in Biotechnology 2018, 38, 1137-1156).

Therefore, as per your recommendation, the following paragraph has been removed:

The crystallographic structure of the SeNPs was analyzed by XRD. The patterns obtained by XRD are interpreted as follows: sharp and strong peaks represent a crystalline structure, whereas low θ-degree broadbands point to the amorphous halos of the particles [48].

Round 2

Reviewer 1 Report

Dear Dr.

the manuscript can be published in present form